# Germline *TP53* Testing in Breast Cancers: Why, When and How?

**DOI:** 10.3390/cancers12123762

**Published:** 2020-12-14

**Authors:** D. Gareth Evans, Emma R. Woodward, Svetlana Bajalica-Lagercrantz, Carla Oliveira, Thierry Frebourg

**Affiliations:** 1Manchester Centre for Genomic Medicine, Division of Evolution and Genomic Sciences, University of Manchester, Manchester M13 9WL, UK; Emma.Woodward@mft.nhs.uk; 2Manchester Centre for Genomic Medicine St Mary’s Hospital, Manchester University Hospitals NHS Foundation Trust, Manchester M13 9WL, UK; 3Hereditary Cancer Unit, Department of Clinical Genetics, Karolinska University Hospital, SE-17176 Stockholm, Sweden; svetlana.lagercrantz@ki.se; 4i3S-Instituto de Investigação e Inovação em Saúde, Universidade do Porto, 4200-135 Porto, Portugal; carlaol@i3s.up.pt; 5Ipatimup-Institute of Molecular Pathology and Immunology of the University of Porto, 4200-135 Porto, Portugal; 6Porto Comprehensive Cancer Center, 4200-072 Porto, Portugal; 7Department of Genetics, Rouen University Hospital, Normandy Centre for Genomic and Personalized Medicine, 76000 Rouen, France; 8Inserm U1245, Normandie University, UNIROUEN, Normandy Centre for Genomic and Personalized Medicine, 76183 Rouen, France

**Keywords:** breast cancer, gene panels, *TP53*, variant, interpretation, Li-Fraumeni syndrome, heritable *TP53*-related cancer syndrome

## Abstract

**Simple Summary:**

*TP53* variants detected in blood represent a main genetic cause of breast cancers occurring before 31 years of age. *TP53* being included in most of the cancer gene panels, patients with breast cancer are offered germline *TP53* testing, independently of the age of tumour onset and familial history. Interpretation of *TP53* variants is remarkably complex, and detection of a germline disease-causing *TP53* variant in a breast cancer patient has drastic medical consequences: radiotherapy contributing to the development of subsequent tumours should be, if possible, avoided. In her family, variant carriers should be offered annual follow-up, including whole-body MRI. Therefore, we consider that, in breast cancer patients, germline *TP53* testing should be performed before treatment and that the decision of *TP53* testing should not be systematic but based on the age of tumour onset, type of breast cancer, personal and familial history of cancer.

**Abstract:**

Germline *TP53* variants represent a main genetic cause of breast cancers before 31 years of age. Development of cancer multi-gene panels has resulted in an exponential increase of germline *TP53* testing in breast cancer patients. Interpretation of *TP53* variants, which are mostly missense, is complex and requires excluding clonal haematopoiesis and circulating tumour DNA. In breast cancer patients harbouring germline disease-causing *TP53* variants, radiotherapy contributing to the development of subsequent tumours should be, if possible, avoided and, within families, annual follow-up including whole-body MRI should be offered to carriers. We consider that, in breast cancer patients, germline *TP53* testing should be performed before treatment and offered systematically only to patients with: (i) invasive breast carcinoma or *ductal carcinoma* in situ (DCIS) before 31; or (ii) bilateral or multifocal or HER2+ invasive breast carcinoma/DCIS or phyllode tumour before 36; or (iii) invasive breast carcinoma before 46 and another *TP53* core tumour (breast cancer, soft-tissue sarcoma, osteosarcoma, central nervous system tumour, adrenocortical carcinoma); or (iv) invasive breast carcinoma before 46 and one first- or second-degree relative with a *TP53* core tumour before 56. In contrast, women presenting with breast cancer after 46, without suggestive personal or familial history, should not be tested for *TP53.*

## 1. Introduction

The development of cancer multi-gene panels has resulted in an exponential increase of *TP53* testing in patients with breast cancers, as *TP53* is included in most of the commercial or academic cancer gene panels. In 1990, heterozygous germline *TP53* variants were shown to be the genetic cause of Li-Fraumeni syndrome (LFS) [1,2,3,4]. LFS is typically characterized by a strong familial clustering of early-onset malignancies and core tumours: soft-tissue sarcomas (STS), osteosarcomas (OS), adrenocortical carcinomas (ACC), central nervous system (CNS) tumours and very early-onset female breast cancers, commonly occurring before 31 years. Thirty years after the characterization of LFS molecular basis, testing children with malignancies or adult females with very early-onset breast cancers without the aspect of family history has shown that familial history of cancer is not mandatory to identify a germline disease-causing *TP53* variant [5,6,7,8,9,10,11,12]. This is explained by the fact that *de novo TP53* variants are not uncommon and penetrance of the variants is incomplete [10,11,12,13]. New criteria, designated the Chompret criteria, have been sequentially updated in order to facilitate the clinical recognition of the syndrome and to cover its clinical heterogeneity [14]. The concept of a heritable *TP53-*related cancer (h*TP53*rc) syndrome has, therefore emerged in order to highlight this clinical diversity [14].

As indicated in Table 1, Chompret criteria include, in particular, (i) the familial aggregation of a patient with breast cancer before 46 years and at least one first- or second-degree relative with a *TP53* core tumour before 56 years, (ii) the development of breast cancer before 46 years and a second tumour and (ii) the occurrence of very early-onset breast cancer, occurring before 31 years.

*TP53* testing is now offered to patients with breast cancers outside these criteria, using cancer gene panels, sometimes via their oncologist or surgeon or even through direct to consumer testing, rather than through genetic counselling services. In this context, we review here the complexity both of *TP53* variant interpretation and cancer risk estimation in variant carriers and the medical consequences of germline disease-causing *TP53* variant identification in breast cancer patients.

## 2. Interpretation of Germline TP53 Variants

When a *TP53* variant is detected from blood in a patient with cancer, two questions should be addressed before considering that the detected variant is the cause of the underlying cancer: (i) Can the detected *TP53* variant be classified as a disease-causing variant? (ii) What is the mutant allele frequency in the blood sample, and is it confined to peripheral blood cells? These questions are clinically critical, considering the consequences for the patients and families of the identification of a germline disease-causing *TP53* variant.

### 2.1. Disease-Causing Variants

Most of the germline disease-causing *TP53* variants are missense and occur within the large central DNA binding domain of the protein [12]. A subclass of these missense variants acts in a dominant-negative fashion resulting in mutant proteins that form tetramers with wild-type p53 and thereby inhibit the transcriptional activity of the wild-type protein [12]. As many missense variants may have no biological impact, interpretation of germline *TP53* variants requires expertise to ensure proper classification, according to the American College of Medical Genetics and Genomics (ACMG)/ Association for Molecular Pathology (AMP) international guidelines: class one, non-pathogenic; class two, likely non-pathogenic; class three, variants of uncertain significance (VUS); class four: likely-pathogenic or class five: pathogenic [15]. As for other genes involved in Mendelian diseases, classification of germline *TP53* variants is based on several arguments including the frequency of the variants in the general population, as recorded in the Genome Aggregation Database [16]; bioinformatics predictions of the variant impact on protein or RNA splicing, using different algorithms; phenotypical and segregation data. Furthermore, interpretation of *TP53* variants also relies on several functional analyses. The first assay commonly used for *TP53* variant interpretation is based on the cloning of human *TP53* cDNAs in yeast expression vectors and the measurement of p53 transcriptional activity in yeast strains containing reporter plasmids including different p53 binding sites [17]. In the yeast assay, p53 variants are classified as functional, not functional, or partially functional according to the level of transcriptional activity quantified using the different reporter plasmids. Two high throughput p53 functional assays have recently been developed in human cancer cell lines. In the first assay, Kotler et al. [18] generated a synthetic library of *TP53* variants located within the p53 DNA-binding domain and quantified their anti-proliferative activity in a p53-null cancer cell line. In this assay, *TP53* variants are categorized as variants retaining wild-type p53 anti-proliferative activity (wild-type *TP53*-like or functional variants) or as variants disrupting this activity (non-functional variants). In the second assay, Giacomelli et al. [19] tested the ability of *TP53* variants generated by saturation mutagenesis (i) to restore the survival of a p53-null cell line exposed to high doses of DNA damaging agents, in order to detect loss of function (LOF) variants and (ii) to induce in p53-wild-type cells resistance to Nutlin-3, in order to detect variants with dominant-negative effect (DNE). The TP53 International Agency for Research on Cancer (IARC) database) has aggregated the results from functional assays and curated both somatic and germline variants [20]. A p53 assay directly performed on blood and based on the quantification of the p53-mediated transcriptional response to DNA damage in the genetic context of patients has recently been developed in order to facilitate the interpretation of variants, which remain of uncertain significance [21]. Specific ACMG/AMP criteria for germline *TP53* variant classification have been defined by a *TP53* variant curation expert panel, under the umbrella of ClinGen. These specific criteria indicate that at least two different functional analyses are required to predict pathogenicity [22].

The complexity of germline *TP53* variant interpretation can be seen by the potential for ‘over-classification’ of *TP53* variants resulting in a frequency of pathogenic or likely-pathogenic variants as high as 1 in 500 in the control database gnomAD [23], whereas a more parsimonious approach provides a more likely 1 in 5000 frequency [24,25]. Over-classification of *TP53* variants has, in particular, resulted in almost certainly overstated risk of colorectal cancer in LFS [26], with one report misclassifying four/six germline variants as pathogenic or likely-pathogenic [27]. As such it can be seen that accurate variant classification is vital when dealing with *TP53* variants identified on germline testing. This process is dynamic as classification should be updated according to the growing knowledge and curated databases. A significant fraction of germline *TP53* variants detected by medical laboratories remains of uncertain significance (class three). Only pathogenic or likely-pathogenic variants (the difference between these two classes being particular subtle for *TP53* and distinction requires specific expertise) should be considered and used in a medical setting. Both types of variants are designated in this manuscript as disease-causing variants.

### 2.2. Mosaic Variants Versus Clonal Haematopoiesis and Circulating Tumour DNA

The implementation of Next-Generation sequencing (NGS) in diagnostic laboratories has greatly improved, thanks to the depth of sequencing, the detection of mutant alleles in a small fraction of genomic DNA extracted from blood. This can be extremely useful in identifying mosaic variants in monogenic conditions [28]. The presence of true mosaic *TP53* alterations should be considered in patients with tumours strongly suggestive of a disease-causing *TP53* variant, such as childhood ACC, choroid plexus carcinoma, breast cancer before 31 years of age or with multiple primary tumours belonging to the *TP53* core tumour spectrum [13].

However, there are two particular pitfalls in *TP53* testing. Whilst true ‘mosaicism’ is not an infrequent occurrence in *TP53* where multiple tissues have low levels of the variant, a more frequent cause of detected low allele frequency is clonal haematopoiesis of indeterminate potential (CHIP) [29,30,31,32]. CHIP corresponds to the expansion of a mutant hematopoietic stem and progenitor cell and is associated with increased risks of haematological neoplasms, including myelodysplastic syndromes and acute myeloid leukaemia. The *TP53* gene is one of the most frequently mutated genes observed in CHIP. CHIP was first reported in patients over 70 years of age but can be detected from 30 years of age. The frequency of CHIP increases with age, tobacco use and exposure to chemo- or radiotherapy. CHIP has recently come to the fore in the context of the exponential development of gene panel testing performed in particular in cancer patients older than 40 years, who smoke or have undergone oncological treatment (e.g., 60 years+ women with breast cancer undergoing panel testing after radiotherapy). Therefore, when a *TP53* variant is detected in a small fraction of NGS reads from genomic blood DNA, it is critical to confirm the presence of the variant in the tumour and in another tissue without mononuclear cells, such as hair follicle or skin biopsy [14]. The detection, in the tumour, of a loss of heterozygosity (LOH) affecting the wild-type *TP53* allele will constitute a strong argument in favour of the causal role of the detected *TP53* variant in tumour development and the reality of the mosaicism. If the variant is not detected within the tumour (best tested on sections without infiltration with lymphocytes or macrophages) or other tissues, clonal haematopoiesis is by far the most likely answer. CHIP should be suspected, in particular, when a *TP53* variant is found in a clinical context not usually associated with germline *TP53* [29,30].

The second pitfall, when mutant alleles are detected in a minor fraction of NGS reads, corresponds to circulating tumour DNA (ctDNA), commonly observed in patients with metastatic cancers. For instance, the detection in a small fraction of NGS reads of a *TP53* variant in the DNA extracted from a patient with metastatic ovarian cancer will not correspond to true mosaicism but very likely to ctDNA [33], considering the very high frequency of somatic *TP53* alterations in these malignancies (>95%). As patients with ovarian cancers are commonly tested using panels, including *TP53*, this situation is becoming frequent.

Therefore, the interpretation of a *TP53* variant detected in a small fraction of reads in a breast cancer patient requires careful analysis integrating the type and stage of the tumour, the age of the patient, and the treatments and will often require complementary genetic investigations performed on other tissues.

## 3. Cancer Risk Associated with Germline Disease-Causing *TP53* Variants

Another area of difficulty concerning *TP53* variant carriers is estimating their cancer risk or penetrance associated with each specific *TP53* variant. The cumulative cancer risk associated with germline disease-causing *TP53* variants was initially calculated mainly from familial cases and was estimated to 73%–100% by age 70, with risks close to 100% in women [34,35,36]. There is nonetheless a clear ascertainment bias leading to an overestimation of disease penetrance when evaluating penetrance from cases. As such, there is no accurate assessment of cancer risk as so few truly prospective studies have been carried out for long enough periods.

What is clear is that, in germline *TP53* disease-causing variant carriers, female breast cancer represents the main cancer risk. Breast cancer risk increases significantly after the second decade, is very high under the age 31 close to 20%–30% on the basis of the Breast cancer RIsk after Diagnostic GEne Sequencing (BRIDGES) study [37], reaches a peak between 25–35 years, and this does tally with estimates from kindreds [38]. This risk drops after 40 years of age based on the relative frequency of *TP53* carriers identified [37,39,40], and cumulative risk reaches a plateau before 60 [35,36,38]. For instance, of 65 known *TP53* carriers with breast cancer in the Genomic Medicine Centre in Manchester 32/65 occurred aged <31 (49%), 24 (37%) occurred aged 31–39, 5 (7.5%) aged 40–44 and only 3/65 (4.5%) over age 45. It is less clear what the overall penetrance is for breast cancer. Although cumulative risk estimates to age 80 for female breast cancer of over 90% have been published [38], these likely suffer from ascertainment and survival bias with many carriers dying from other malignancies at young ages. There is clearly a very high rate of contralateral breast cancer approaching 4%–7% annually and significantly higher than *BRCA1* or *BRCA2* in those diagnosed aged <35 [41].

It is also clear that penetrance of germline disease-causing *TP53* variants is not only incomplete but variable. The molecular bases of the penetrance variability remain to be characterized. One factor explaining, at least in part, this variability is the type of the variant itself. As explained previously, many missense variants are classified as dominant-negative due to their ability to complex and reduce the transcriptional activity of wild-type p53 protein, which normally acts as a tetramer. These dominant-negative missense *TP53* variants are predominantly detected in families with childhood malignancies and are generally more penetrant in childhood than the other types of alterations. In contrast, typical loss of function variants (frameshift or nonsense variants, splicing variants, large genomic rearrangements), and non-dominant-negative missense variants, are predominantly identified in families with mostly adult cancers and appear to have a lower disease penetrance in childhood [12]. A particularly important example of a low penetrant, but still pathogenic variant, is the non-dominant-negative missense p.Arg337His variant, present in 0.3% of the Southern Brazilian population [42,43,44,45]. The lower penetrance of these variants may be explained by a less drastic impact of p53 transcriptional activity, as compared to the dominant missense variants [46]. However, it should be highlighted here that penetrance is a dynamic process that should be expressed as a curve according to age. Therefore, germline disease-causing *TP53* variants designated as low penetrant in childhood, such as the canonical p.Arg337His variant, can nevertheless be associated with a high cumulative breast cancer risk up to 63% in adults as shown by recent results obtained in the Brazilian population (Maria-Isabel Achatz, personal communication). The variability in the age of tumour-onset among relatives harbouring the same germline *TP53* variant clearly shows that the penetrance also depends from modifier factors that can be either genetic or environmental and their identification in the future would allow a more personalized clinical management of *TP53* variant carriers.

## 4. Features of Breast Tumours in *TP53* Variant Carriers

### 4.1. Age of Tumour-Onset

Several studies [47] have shown that, independently of the familial history, germline disease-causing *TP53* alterations are identified in between 3.8% and 7.7% in females with breast carcinoma before 31 years of age, but a Dutch study recently reported a lower rate of 2% [48]. After 30 years, the rate of germline *TP53* drops sharply, and germline *TP53* variants rarely cause familial breast cancer unexplained by *BRCA1* or *BRCA2* [39,49,50]. The very low rate of germline *TP53* mutation detected in breast carcinoma presenting after age 30 years has recently been validated in the BRIDGES study, including 60,466 cases and 53,461 controls [37]. In the BRIDGES study, 4/346 patients < 30 years had germline truncating *TP53* variants compared to 2/53,461 controls yielding an odds ratio (OR) of 309, contrasting with only 3/60,120 > 30 years and an OR of 1.33 (Easton D, personal communication).

### 4.2. Histopathologic Features

The vast majority (above 90%) of female breast cancers observed in germline disease-causing *TP53* variant carriers, corresponds to invasive ductal carcinomas of no special type. The remaining cases correspond to invasive lobular carcinoma or invasive ductal/lobular carcinoma [9,40,51,52,53,54]. In contrast to *BRCA1/BRCA2* variant carriers, a high frequency of *ductal carcinoma* in situ (DCIS) is observed, with estimates up to 25% [52,53], and it is likely that this fraction will increase with systematic annual breast MRI in carriers. A high nuclear grade is observed in the majority of the invasive carcinoma and DCIS cases associated with germline *TP53* variants, and most of the invasive carcinoma are of mSBR (modified Scarff–Bloom–Richardson) grade three [52,53]. Most of the breast carcinoma exhibits HER2 amplification and overexpression, as 60%–83% show HER2 positivity [9,40,49,51,52,53,54] with more than 40% showing ER co-expression. However, using HER2+ as the only argument to test over 30 years of age identifies very few carriers [40]. Based on this single criterion, only 2/82 breast cancer patients <40 (2.5%) had a germline *TP53* pathogenic variant, 1/132 aged ≤40 and among women without a known family history consistent with Chompret criteria, *TP53* variants were found in only 1/195 (0.5%). Nonetheless, some weight of HER2 expression can be considered in association with other arguments suggestive of genetic determinism, such as the early age of onset, the high nuclear grade, the histological type and tumour multifocality.

Beside breast carcinomas, phyllodes tumours of the breast, which are rare mesenchymal tumours with differential malignant potential, are also strongly suggestive of a germline *TP53* alteration, when they occur in patients before 36 years of age. The association between germline *TP53* alterations and early-onset phyllodes tumours has been documented since 2001 by several articles [55,56] and observations from different teams, and a recent study reported a mutation detection rate up to 10% [57]. Most of the reported phyllodes tumour cases associated with germline *TP53* variants are malignant [55,56].

## 5. Treatment-Related Risks in *TP53* Variant Carriers

Germline pathogenic *TP53* variant carriers have a very high risk of subsequent primary tumours, which may occur in >40% of *TP53* carriers [12,35]. Subsequent primary tumours often develop after treatment of *TP53* carriers with radiation and/or genotoxic chemotherapy. The contribution of radiotherapy and conventional chemotherapy to the development of subsequent primary tumours came initially from multiple observations in kindreds and was strongly suggested by the key role of p53 in transcriptional response to DNA damage [12]. Functional studies performed ex vivo on lymphocytes have shown that radiotherapy and conventional chemotherapies, except mitotic spindle poisons, induce the p53-mediated transcriptional response to DNA damage and that this response is altered in non-malignant cells from germline disease-causing *TP53* carriers [58]. Furthermore, exposure of mice, harbouring a germline alteration of one *TP53* allele, to radiotherapy or genotoxic chemotherapies drastically increases tumour development risk [58]. All these data show that, unfortunately, radiotherapy and genotoxic chemotherapies used for the treatment of first cancer contribute to the development of subsequent primary tumours in *TP53* variant carriers. Our conviction of this risk is re-enforced by the growing number of observations of sarcomas occurring in the radiotherapy field, in *TP53* variant carrier.

Therefore, in patients with breast cancers, if the age of tumour onset, personal or familial history is suggestive of a germline pathogenic *TP53* variant, testing for disease-causing *TP53* variants should be carried out before starting treatment, in order to prioritize radical surgical treatment and discuss, in the framework of a multi-disciplinary team, the balance between the risk of recurrence in the absence of chemo- or radiotherapy and the risk of secondary primary tumour induced by the treatment [14]. For instance, if a disease-causing *TP53* variant is identified in a young woman with invasive, T1N0 breast cancer, this will mean strongly advising for mastectomy rather than breast-conserving surgery and radiotherapy and given the high contralateral risk the option of bilateral mastectomy [14,41]. In breast cancer patients, harbouring a germline disease-causing *TP53* variant and treated by radiotherapy, the development of sarcomas in the radiotherapy field becomes a growing issue, and their early detection should constitute a priority during their annual follow-up. In this context, the detection in a breast cancer patient of a germline *TP53* variant of unknown significance is particularly challenging and should lead to an updated interpretation by an expert laboratory.

## 6. Surveillance Protocols in Carriers of Germline Disease-Causing *TP53* Variants

Guidelines for surveillance of *TP53* disease-causing variant carriers, based on the Toronto protocol [59], have recently been published [14,60,61]. These protocols exhibit some difference, but they all include abdominal ultrasound every 3–6 months, annual whole-body MRI (WBMRI) and annual brain MRI (the first with gadolinium enhancement) from the first year of life. Additionally, female carriers should undergo annual breast MRI from the age of 20 onwards. The option of risk-reducing mastectomy may be discussed on a case-by-case basis [14,60]. Several international studies, mostly performed without gadolinium-based contrast agents (GBCAs), have confirmed the efficiency of WBMRI, with an overall estimated detection rate of 7% for new and localized primary cancers on a first prevalent screen [59,62,63,64,65,66,67,68]. Given concerns over the accumulation of gadolinium, it is advised that only initial scans should have this on WBMRI and brain imaging [14] although this is more problematic to avoid for breast imaging. Table 2 presents the recommended follow-up according to the guidelines elaborated by the European Network, Genetic tumour risk syndromes (GENTURIS) [14].

## 7. Impact of a Germline Disease-Causing *TP53* Variant on Genetic Counselling

Detection of a germline disease-causing *TP53* variant in a breast cancer patient has not only major medical impact for herself but for their relatives. The variability of the age of tumour-onset observed in *TP53* variant carriers, even within a family, complicates genetic counselling. Indeed, in carriers, the benefit of the annual surveillance program should be evaluated in the perspective of *TP53* variant incomplete penetrance and the psychological burden induced by an intensive annual medical follow-up starting in childhood. The first international guidelines, based on the Toronto protocol [59] and published in 2017 [60], recommended to offer systematically the same surveillance protocol and therefore pre-symptomatic testing in childhood since the first year of age. However, the increasing number of germline *TP53* variants detected in cancer patients without familial history, the incomplete and variable penetrance of the variants, the observation that only a fraction of variant carriers have access to this protocol, especially in US, lead us to propose a more stratified protocol in the framework of the European Reference Network GENTURIS, in order to help decision making [14]. For instance, when a germline *TP53* pathogenic variant is detected in a patient with a breast cancer at 29 years of age without familial history, or in a 44-year breast cancer patient whose mother developed a sarcoma at 51, should we systematically offer pre-symptomatic testing in childhood since the first year of age? In the recently published European guideline [14], we recommend systematic pre-symptomatic testing and the intensive protocol in childhood from birth, under the following conditions: childhood cancers have been observed within the family, or this variant has already been detected in other families with childhood cancers, or this variant corresponds to a dominant-negative missense variant. If these conditions are not observed, we do not systematically recommend pre-symptomatic testing and complete annual medical follow-up since the first year of age. We think nevertheless that testing children in families with only early-onset adult cancers can be considered, but only after careful discussion with the parents in order to address the burden, and uncertain benefits, of surveillance in childhood [14]. The complexity of genetic counselling in families with germline *TP53* variant carriers constitutes an additional argument highlighting the need to address the question of *TP53* testing in the framework of a multi-disciplinary team.

## 8. Psychological Considerations

Given the variability, potential high penetrance and early onset of disease associated with *TP53* pathogenic variants, individuals with this diagnosis may need additional psychosocial support to deal with distress and bereavement linked to cancer diagnoses in them and/or their family [69]. In contrast to the detection of a causal germline alteration in the other main breast cancer genes such as *BRCA1*, *BRCA2* or *PALB2,* identification of a germline disease-causing *TP53* variant in a patient with breast cancer may have an impact in her children under 18 years of age. Furthermore, genetic counselling, with further information on financial/insurance implications is necessary to ensure an informed decision has been made with regard to surveillance adherence.

## 9. Conclusions

Testing women with breast cancer for *TP53* variants is not a straightforward decision. From the above, it can be seen that women identified as carriers of disease-causing *TP53* variants have a lifelong indication for a heavy surveillance protocol (Table 2). The identification of a disease-causing *TP53* variants variant will also affect their treatment with a decision on whether to opt not just for unilateral but bilateral mastectomy and will have important consequences, in terms of genetic counselling, sometimes in children before 18 years of age. It is clearly vital that any testing of *TP53* results in a robust analysis of the variant by a laboratory with expert experience and using the most up to date criteria [22]. A complex situation is represented by the detection of a class three variant in a patient with breast cancer. If the variant remains a class 3 despite expert analysis and if the clinical presentation is strongly suggestive of a germline disease-causing *TP53* variant, we think that it is safer to ensure in the patient the recommended follow-up with a regular update of the variant classification. In contrast, without certainty of the variant disease-causing role, prophylactic mastectomy should not be considered, and pre-symptomatic testing in relatives should not be performed, according to the rules of medical genetics. The most complicated issue in this situation remains the question of radiotherapy which should be discussed, case by case, by a multi-disciplinary team, including expertise in *TP53*.

As indicated in Table 3, the decision of *TP53* testing in a patient with breast cancer should be based first on the updated Chompret criteria (Table 1), which constitute a formal indication.

Besides the Chompret criteria, in breast cancer patients, this decision should also integrate the combination of several parameters including age of tumour-onset, histopathological characteristics, multifocal type, and familial history of cancers, which justifies extending the Chompret criteria in breast cancer patients (Table 3). Given (i) the very low rate of clearly disease-causing *TP53* variants aged in women with breast cancer aged > 46, (ii) the reasonably high chance of a variant of uncertain significance that could be misclassified and (iii) the increasing likelihood of clonal haematopoiesis, we consider that women presenting with breast cancer after 46 years, without personal or familial history, fulfilling the “Chompret Criteria” should not be tested using a breast panel that includes *TP53* (Table 3). We also recommend that any patient presenting with isolated breast cancer not fulfilling the criteria presented in Table 3 and in whom a *TP53* variant has been identified, should be referred to an expert genetics service that can interpret its true meaning and to a multi-disciplinary team for discussion [14]. This will ensure validation that will classify the variant accurately according to ACMG/AMP criteria and distinguish true h*TP53*rc from CHIP and erroneous classification as h*TP53*rc.

## Figures and Tables

**Table 1 cancers-12-03762-t001:** Updated Chompret Criteria for Germline *TP53* Testing [14].

Presentation	Criteria
Familial presentation	Patient with a *TP53* core tumour (breast cancer, soft-tissue sarcoma, osteosarcoma, central nervous system tumour, adrenocortical carcinoma) before 46 years AND at least one first- or second-degree relative with a core tumour before 56 years
Multiple primitive tumours	Patient with multiple tumours, including 2 *TP53* core tumours, the first of which occurred before 46 years, irrespective of family history
Rare tumours	Patient with adrenocortical carcinoma, choroid plexus carcinoma, or rhabdomyosarcoma of embryonal anaplastic subtype, irrespective of family history
Very early-onset breast cancer	Breast cancer before 31 years, irrespective of family history

**Table 2 cancers-12-03762-t002:** Surveillance Protocol in Carriers of Germline disease-causing *TP53* Variants (from reference 14).

Exam	Periodicity	Age to Start	Age to End	Condition
Clinical examination	Every six months	Birth	17 years	
	Annual	18 years	-	
Whole-Body MRI without gadolinium enhancement	Annual	Birth	-	*TP53* variant conferring high cancer risk in childhood *
		18 years	-	
Breast MRI	Annual	20 years	Until 65 years	
Brain MRI **	Annual	Birth	18 years	*TP53* variant conferring high cancer risk in childhood
		18 years	Until 50 years	
Abdominal ultrasound	Every six months	Birth	Until 18 years	
Urine steroids	Every six months	Birth	Until 18 years	When abdominal ultrasound does not allow a proper imaging of the adrenal glands
Colonoscopy	Every five years	18 years	-	Only if the carrier received abdominal radiotherapy for the treatment of a previous cancer or if there is a familial history of colorectal tumours suggestive of an increased genetic risk

* A germline disease-causing *TP53* variant should be considered as “high risk” in childhood if the index case has developed a childhood cancer; or childhood cancers have been observed within the family, or this variant has already been detected in other families with childhood cancers, or this variant corresponds to a dominant-negative missense variant. ** The first scan should be conducted with I.V. Gadolinium enhancement; in children, brain MRI should alternate with the Whole-Body MRI, so that the brain is imaged at least every six months.

**Table 3 cancers-12-03762-t003:** Germline *TP53* testing in Women with Breast Tumour.

Age of Breast Tumour Onset	Presentation
*When TP53 testing should systematically be performed*	
Before 31	Invasive breast carcinoma ^a^ or *ductal carcinoma* in situ (DCIS)
Before 36	– *Bilateral* invasive breast carcinoma or DCIS – or *Multifocal* invasive breast carcinoma or DCIS – or *HER2+* invasive breast carcinoma or DCIS – Phyllode tumour
Before 46	– Invasive breast carcinoma *and* a second *TP53* core tumour in the patient ^a^ – or invasive breast carcinoma *and* one first- or second-degree relative with a *TP53* core tumour before 56 years ^a,b^
*When TP53 testing may be offered*	
Before 46	– *Bilateral* invasive breast carcinoma – or *HER2+* invasive breast carcinoma *and* a familial history of HER2+ breast cancer
*When TP53 testing should not be performed*	
After 46	No previous *TP53* core tumour before 46 and no familial history fulfilling Chompret criteria

^a^ Chompret criteria (14), ^b^
*TP53* core tumour: breast cancer, soft-tissue sarcoma, osteosarcoma, central nervous system tumour, adrenocortical carcinoma. DCIS: *ductal carcinoma* in situ.

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
