# Peer review of "Germline TP53 Testing in Breast Cancers: Why, When and How?"

_cancers, 2020, doi:10.3390/cancers12123762_

Round 1
Reviewer 1 Report
This manuscript is a very complete review about why, when and how TP53 genetic testing in breast cancer patients must be performed. Besides its high relevance in the field of hereditary cancer and genetic susceptibility to this disease, the main interest of the manuscript resides in its potential to provide simple and practical considerations for clinical managing TP53 carriers.
The principal authors are very well experienced researchers with remarkable publications in the field and high H-indexes. In addition, the manuscript is very well presented and clear. It includes many considerations related to the topic in hand, all of them written using scientific language but keeping it light and very understandable so it cannot be misinterpreted by professionals. I have particularly appreciated the critical review regarding genetic testing in children and the addiction of a psychological consideration section, which is not so common in this kind of reviews.
I do not have major concerns about this submission or particular criticisms to raise. However, I would like to report to the authors some suggestions that in my opinion might improve the quality and readability of the paper and a few minor points.
The authors state that only class 4 and 5 variants should be considered and used in a medical setting. However, what happens with class 3 variants? Do the authors think that patients carrying them should be managed as if they carried class 1 or 2 variants? Maybe some clarification on this point would be interesting and appropriated.
Minor points.
The abbreviation form for ductal carcinoma in situ (DCIS) is described as such in line 228. However, DCIS appears prior that in the abstract. If the number of words of that section allows it, I think it may be beneficial to introduce the term there.
Please, double check page numbering. It looks as if there is a formatting issue.
Reviewer 2 Report
This review paper describes germline p53 testing in breast cancer patients. I only have the following minor comments:
- Line 227: change "ductal carcinomas" to "invasive ductal carcinomas of no special type."
- 4.2. Histopathologic Features: a) please comment if there is any difference in tumor grade in TP53 mutation carriers versus the general population; b) please specify what other special subtypes of invasive breast carcinoma have been described in TP53 mutation carriers; b) please comment on phyllodes tumor grade in TP53 mutation carriers, ie are these tumors usually malignant as opposed to benign or borderline?
- Table 3: define DCIS at the end of the table.
